
# Coastal ocean forecasting with an unstructured-grid model in the Southern Adriatic Northern Ionian Sea

Ivan Federico[1], Nadia Pinardi[1,2,3], Giovanni Coppini[1], Paolo Oddo[2,4], Rita Lecci[1], and Michele Mossa[5]

[1]Centro Euro-Mediterraneo sui Cambiamenti Climatici – Ocean Predictions and Applications, via Augusto Imperatore 16, 73100 Lecce, Italy
[2]Istituto Nazionale di Geofisica e Vulcanologia, Via Donato Creti 12, 40100 Bologna, Italy
[3]Universitá degli Studi di Bologna, viale Berti-Pichat, 40126 Bologna, Italy
[4]Now at NATO Science and Technology Organisation – Centre for Maritime Research and Experimentation, Viale San Bartolomeo 400, 19126 La Spezia, Italy
[5] Dipartimento di Ingegneria Civile, Ambientale, del Territorio, Edile e di Chimica, Politecnico di Bari, Via E. Orabona 4, 70125 Bari, Italy

*Correspondence to:* Ivan Federico (ivan.federico@cmcc.it)

**Abstract.** SANIFS (Southern Adriatic Northern Ionian coastal Forecasting System) is a coastal-ocean operational system based on the unstructured-grid finite-element three-dimensional hydrodynamic SHYFEM model and providing short-term forecasts. The operational chain is based on a downscaling approach starting from the large-scale system for the entire Mediterranean basin (MFS, Mediterranean Forecasting system), which provides initial and boundary condition fields to the nested system.

The model is configured to provide hydrodynamics and active tracer forecasts both in open ocean and coastal waters of South-eastern Italy using a variable horizontal resolution from 3-4 km in the open sea to 500-50 m in the coastal areas.

Given that the coastal fields are driven by a combination of both local/coastal and deep ocean forcings propagating along the shelf, the performance of SANIFS was verified both in forecast and simulation mode, first (i) at the large and shelf-coastal scales by comparing with a large scale survey CTD in the Gulf of Taranto and then (ii) at the coastal-harbour scale (Mar Grande

of Taranto) by comparison with CTD, ADCP and tide gauge data.

Sensitivity tests were performed on initialization conditions (mainly focused on spin-up procedures) and on surface boundary conditions by assessing the reliability of two alternative datasets at different horizontal resolution (12.5 and 6.5 km).

The SANIFS forecasts at a lead-time of one day were compared with the MFS forecasts highlighting that SANIFS is able to retain the large scale dynamics of MFS. Those get correctly propagated to the shelf-coastal scale improving the forecast

accuracy (+17% for temperature and +6% for salinity compared to MFS). Moreover the added value of SANIFS was assessed at coastal-harbour scale, which is not covered by the coarse resolution of MFS, where the SANIFS forecasted fields well reproduced the observations (temperature RMSE equal to 0.11 °C).

Furthermore SANIFS simulations were compared with hourly time-series of temperature, sea level and velocity measured at the coastal-harbour scale showing a good agreement. Simulations in the Gulf of Taranto described a circulation mainly

characterized by an anticyclonic gyre with the presence of cyclonic vortexes in shelf-coastal areas. A surface water inflow from open sea to Mar Grande characterizes the coastal-harbour scale.



# 1 Introduction

Many human activities are concentrated in coastal areas where traditional resource-based activities, such as coastal fisheries and aquaculture, coexist with urban development, port traffic and tourism. The management of such complex area, at the interface between land and ocean environments, requires numerical modelling and predictive capabilities that are now possible due to the availability of large-scale ocean forecasts and analyses used to initialize coastal models that downscale and increase the accuracy of the forecasts near the coasts (Pinardi et al. (2003); Pinardi and Coppini (2010)). Furthermore ocean operational forecasting contributes greatly to the understanding of ocean dynamics, as well as providing an efficient support tool for marine environmental management (Oddo et al. (2006); Robinson and Sellschopp (2002)).

The main objective of the present work is to highlight how downscaling could improve the simulation of the flow field going from typical open-ocean scales of the order of several km to the coastal (and harbour) scales of tens to hundreds of meters. Two methodologies were adopted: 3D unstructured-grid modelling and a downscaling approach that uses open ocean fields as initial and lateral boundary conditions.

Classical ocean models are based on finite differences schemes on Cartesian grids (Griffies et al. (2000)). It is only in the last decade that unstructured meshes methods have been used more intensively in ocean and coastal modelling, adopting both finite elements (e.g. Umgiesser et al. (2004); Danilov et al. (2004); Walters (2005); Le Bars et al. (2010)) and finite volume (e.g. Casulli and Zanolli (2000); Chen et al. (2003); Ham et al. (2005); Fringer et al. (2006)) discretization. Unstructured grid models are used for coastal modelling, exploiting their efficiency at handling complex coastlines while not neglecting large-scale processes.

Downscaling techniques are the preferred methodology to propagate open-ocean dynamics into higher resolution nested models through initial and boundary conditions (Kourafalou et al. (2015)). Here the downscaling was performed from the Mediterranean scale through a one-way nesting approach with the Mediterranean Forecasting System (MFS) numerical model (Oddo et al. (2014)). The nested unstructured-grid system was built on a resolution higher than MFS both horizontally and vertically over the entire open ocean and coastal domain.

Unstructured-grid and downscaling approaches were incorporated to build the very-high resolution operational system SAN-IFS (Southern Adriatic Northern Ionian coastal Forecasting System) which is constantly under development and which produces short-term forecasts.

The specific coastal area studied in this paper is the Southern Adriatic Northern Ionian (SANI) area (Fig. 1), with a particular focus on the Gulf of Taranto and Mar Grande (Fig. 2). The Southern Adriatic sea extends approximately southward along the latitude of 42°N to the threshold of the Strait of Otranto, and has a maximum depth of 1270 m. An exchange of waters with the Ionian Sea occurs at the Strait of Otranto approximately at 40°N. The Northern Ionian Sea extends south of 38°N and has a steeper continental slope than the Adriatic basin. The offshore maximum depth is 3500-3700 m.

The Gulf of Taranto is situated in the north-western Ionian Sea and is approximately delimited in open sea by the line connecting Cape Santa Maria di Leuca (in Apulia) and Cape Colonna (see Fig. 1). A complex water mass circulation (Sellschopp and Álvarez (2003)) with high seasonal variability (Milligan and Cattaneo (2007)) characterizes the Gulf of Taranto basin.





Oceanographic studies (Poulain (2001); Bignami et al. (2007); Turchetto et al. (2007); Grauel and Bernasconi (2010); Oddo and Guarnieri (2011)) based on interannual simulations describe two main Taranto Gulf surface circulation structures: (i) a cyclonic gyre encompassing the Western Adriatic Coastal Current (WACC) flowing around Apulia into the Taranto Gulf from the northern Adriatic Sea, and (ii) an anticyclonic circulation connected to the Ionian Surface Waters (ISW) flow entering from

the central Ionian Sea. In addition, studies on the coastal current in the inner part of the Gulf of Taranto (De Serio and Mossa (2015a)) show a coastal flow supporting the anticyclonic gyre structure. In the Mar Grande of Taranto (Fig. 2b) the circulation consists of along-shore currents, which follow the direction of the wind and are modulated by the tidal forcing (Scroccaro et al. (2004)) with a tidal range of about 20 cm (Umgiesser et al. (2014)).

A regional cruise, called MREA14 (Maritime Rapid Environmental Assessment in 2014), was carried out in the Gulf of

Taranto in October 2014 in order to provide data for validating the SANIFS downscaled simulations. Thus the SANIFS system has been fully validated at least for the time period of the available data, both at regional and coastal scales.

The paper is organized as follows. In Section 2 the design and implementation of the forecasting system are introduced. Section 3 reports the sensitivity experiments and the validation of the SANIFS system. Section 4 discusses the circulation structures emerging from the SANIFS system during the validation exercise, and concluding remarks are provided in Section

15   5.

## 2   The forecasting system: definition and implementation

The SANIFS methodology is based on the high resolution model re-initialization every day, similarly to the short term limited area atmospheric modelling practice (Mesinger et al. (1988)). In this section we report the main model settings, the surface boundary conditions, lateral boundary conditions and the operational configuration. Those are summarized in Table 1.

### 2.1   Model settings

The SANIFS forecasting system is based on the SHYFEM model which is a 3D finite element hydrodynamic model (Umgiesser et al. (2004); Cucco and Umgiesser (2006)) solving the Navier-Stokes equations by applying hydrostatic and Boussinesq approximations. The unstructured grid is Arakawa B with triangular meshes (Bellafiore and Umgiesser (2010); Ferrarin et al. (2013)), which provides an accurate description of irregular coastal boundaries.

The scalars are computed at grid nodes, whereas velocity vectors are calculated at the center of each element. Vertically a z-layer discretization is applied and most variables are computed in the center of each layer, whereas stress terms and vertical velocities are solved at the layer interfaces (Bellafiore and Umgiesser (2010)).

In the coastal waters of the eastern Italian coastlines, the model has a high spatial resolution, generally reaching an element size of 500m, with further refinements in specific areas (e.g. Mar Grande of Taranto, Fig. 2b) where the resolution reaches 50m.

In the open ocean areas, the horizontal resolution is approximately 3-4 km with respect to the 6-7 km of the parent model.



The SANIFS bathymetry (Fig. 1 and Fig. 2b) was derived from the Digital Bathymetric Data Base Variable Resolutions (DBDB1) at 1' resolution for the Mediterranean basin and integrated with higher resolution bathymetry for coastal areas in the Gulf of Taranto.

The vertical discretization has 99 levels. This is appropriate for solving the field both in coastal and open-sea areas. The vertical spacing is $2m$ until $40m$ from the sea surface, and the resolution is then progressively (stepwise) increased down to the bottom.

A non-linear bottom parameterisation assuming a quadratic bottom friction was imposed. The friction coefficient was expressed as $R = C_D\sqrt{u^2 + v^2 + e_b}$ where, $u$ and $v$ are the horizontal velocities, $C_D$ is the drag coefficient calculated by a logarithmic formulation (Maraldi et al. (2013)) and $e_b$ is the bottom turbulent kinetic energy due to tides, internal waves breaking and other short time scale currents. Following the Treguier's 1992 (Treguier (1992)) experiment and the MFS settings, $e_b$ was set to a value of $2.5 \cdot 10^{-3}\ m^2/s^2$.

A local Richardson number dependent formulation was applied for the vertical momentum and tracer eddy coefficients with a specific constraint in the mixing layer (Lermusiaux (2001)). Using a scheme similar to Pacanowski and Philander (1981), the calculation of eddy viscosities and diffusivities are based on the Richardson number $Ri = N^2/(\partial \bar{U}/\partial z)^2$ where $N^2$ is Brunt-Vaisälä frequency and $\bar{U}(x,y)$ the velocity field.

If $Ri(x,y,z,t) \geq 0$, the eddy viscosity and diffusivity are set to $A_v = A_v^b + (v_0)/(1+5Ri)^2$ and $K_v = K_v^b + (v_0)/(1+5Ri)^3$. Otherwise if $Ri(x,y,z,t) < 0$ a convective adjustment is adopted ($A_v = 5 \cdot 10^{-3}\ m^2/s$ and $K_v = 5 \cdot 10^{-3}\ m^2/s$).

The background molecular coefficients are $A_v^b = 10^{-6}\ m^2/s$ and $K_v^b = 10^{-7}\ m^2/s$. The shear eddy viscosity is $\nu_0 = 5 \cdot 10^{-3}$ $m^2/s$. An enhancement in the mixing layer (Lermusiaux (2001)) was adopted to transfer and dissipate the wind stress and the buoyancy fluxes. The vertical eddy coefficients within the Ekman depth $h_e(x,y,t)$ are set to empirical values calibrated for region and season $A_v^e = 1.5 \cdot 10^{-3}\ m^2/s$ and $K_v^e = 5 \cdot 10^{-4}\ m^2/s$). The Ekman depth is calculated as a function of turbulent friction velocity $u^*(x,y) = \sqrt{|\tau|/\rho_0}$ through the relationship $h_e = E_k u^*/f_0$, where $\tau$ is the wind stress vector, $\rho_0$ the reference density, $f_0$ is the Coriolis factor, and $E_k$ an empirical coefficient set to 0.7.

## 2.2 Surface boundary conditions

Four basic surface boundary conditions are used:

1. for temperature, the air-sea heat flux is parameterized by bulk formulas described in Pettenuzzo et al. (2010);

2. for momentum, surface stress is computed with the wind drag coefficient according to Hellermann and Rosenstein (1983);

3. for the free surface, a water flux is used containing evaporation minus precipitation and runoff;

4. for salinity, the turbulent salt flux is set equal to the product of the water flux and the surface salinity.

Twenty monthly mean climatological river runoffs were considered:



- Italian Ionian sea rivers: Basento, Bradano, Crati, Sinni, Agri, Neto;

- Italian Adriatic sea rivers: Fortore, Cervaro, Ofanto;

- Greek Ionian sea rivers: Arachthos, Thyamis;

- Albania-Montenegro-Croatia Adriatic sea rivers: Vijose, Seman, Shkumbi, Erzen, Ishm, Mat, Drin, Buna/Bojana, Neretva.

River inflow surface salinity values were fixed to a constant value of 15 psu next to the river mouths, following the sensitivity tests carried out with MFS parent model in the shelf areas close to river outlets.

Two alternative atmospheric forcing datasets were used in order to set up two SANIFS operational configurations (see Section 2.4), one forced via ECMWF (European Centre for Medium Weather Forecasts) products with 6h frequency and 12.5km horizontal resolution, the other via COSMOME (based on COSMO model and operated by the Italian National Center of Meteorology and Climatology, CNMCA) products with 3h frequency and 6.5km horizontal resolution.

The atmospheric forcing fields used from the two datasets are: 2m air temperature (T2M), 2m dew point temperature (D2M), total cloud cover (TCC), mean sea level atmospheric pressure (MSL), meridional and zonal 10m wind components (U10M and V10M). In the configuration forced by ECMWF data, the total precipitation (TP) rate data is extracted from the CMAP (CPC, Climate Prediction Center, Merged Analysis of Precipitation) monthly dataset with a horizontal resolution of 2.5°x2.5°. In the configuration forced by COSMOME data, TP derives from the operational COSMOME dataset and is identified as a sum of large-scale precipitation (LSP) and convective precipitation (CP).

Finally the atmospheric forcing fields were corrected by land contaminated points following Kara et al (2007) and horizontally interpolated at each ocean grid node by means of Cressman's interpolation technique (Cressman (1959)).

## 2.3 Lateral open boundary conditions

SANIFS is nested to MFS through the two lateral open boundaries (Fig. 1) located at the southern (horizontal boundary) and northern (oblique boundary) parts of the domain. The current MFS implementation is based on NEMO (Nucleus for European Modelling of the Ocean, Madec (2008) finite-difference code with a horizontal resolution of 1/16 of a degree (6-7 km approximately) and 72 unevenly spaced vertical levels. The forecasting system is provided by a data assimilation system based on the 3DVAR scheme developed by Dobricic and Pinardi (2008).

The scalar MFS fields (non-tidal sea surface height, temperature and salinity) are imposed at the SANIFS boundary nodes, whereas the MFS total velocities are specified only in the baricenter of the triangular elements with two nodes attached to the boundaries. The tidal elevation derived from the OTPS (Oregon State University Tidal Prediction Software (Egbert and Erofeeva (2002)) tidal model are prescribed at each boundary nodes. Eight of the most significant constituents are considered: M2, S2, N2, K2, K1, O1, P1, Q1.



## 2.4 The operational configuration

The operational chain provides 3-day forecasts for the two configurations: (i) SANIFS-ECMWF forced via ECMWF atmospheric data, and (ii) SANIFS-COSMOME forced via COSMOME atmospheric data.

The daily forecast cycle for the two configurations is reported in Fig. 3a. With j as the current day, the initializing fields
(taken from the MFS simulation products) of the SANIFS forecast procedure are imposed at 12:00 a.m. of two days backward (j-2) as the instantaneous fields. The SANIFS forecasting run exploits the MFS simulations (for j-2 and j-1) and the MFS forecasts (for j+1, j+2 and j+3) at the lateral open boundary, while the surface boundary conditions run over the ECMWF and COSMOME analysis (j-2 and j-1) and ECMWF and COSMOME forecasts (j+1, j+2, j+3).

The forecast is prepared and run automatically. The operational chain is activated as soon as the atmospheric forcings are
available. The technical procedures (Fig. 3b) through scripts and codes for computing the forecast fields can be summarized in the pre-processing of input data, model run and post-processing of the output model.

## 3 Sensitivity experiments and validation

### 3.1 The validation data set

Three cruises (MREA14) were organized together with data acquisition in the Gulf of Taranto and Mar Grande (Pinardi et al.
(2016), in this issue). Data available for the validation are: (i) temperature and salinity profiles from CTD stations (Fig. 4) and (ii) hourly temperature measurements, sea level and currents at a fixed station (P1 station in Fig. 4, and described in De Serio and Mossa (2015b)).

The first CTD survey was carried out between 1-3 October 2014 and CTDs were acquired at large scales (labelled LS1 in Fig. 4). The second set of surveys was carried between 5-8 of October 2014 at the shelf-coastal (labelled SCS in Fig. 4) and at
the coastal-harbour scale (labelled CHS in Fig. 4) in Mar Grande. The last, large scale survey (labelled LS2 in Fig. 4) completes the mapping of the circulation between October 8 and 11, 2014.

The main purpose of the large scale surveys was to identify the main thermohaline structures in the Gulf of Taranto and to provide an initialization (LS1) and forecast verification survey (LS2). The SCS and CHS surveys focused on the coastal structures and the water exchange between the open sea and Mar Grande.

### 3.2 Initialization procedure and spin-up time

Limited area ocean models may require a spin-up time to produce dynamically adjusted fields after initialization from the interpolation of coarser ocean model fields (Simoncelli et al. (2011)). Two main issues are addressed here: (i) the dynamical adjustment at the coasts where coarser ocean fields are extrapolated from the initial conditions, and (ii) the sensitivity of the initialization to a different number of dynamical fields.





All the experiments were initialized from daily mean fields produced by the MFS parent model. An extrapolation procedure (De Dominicis et al. (2013)) is used to prevent the presence of missing values interpolating of the oceanic fields over the new higher resolution grid.

Two experiments are used: the first (SANIFS-v0) considers the nested model initialized only with temperature and salinity

fields from MFS. In the second one (SANIFS-v1), the initialization uses temperature and salinity, sea level and currents. Each of these experiments is started at different times in the past with respect to a target initial forecast time. The number of days in the past with respect to the target initial forecast day is called the spin-up time and our aim was to test how long the spin-up needed to be in order to get closer to the observations in the initial condition. The target forecast initial conditions day is October 8 2014 and the spin-up time was evaluated up to five days in the past.

A spin-up time indicator is defined as the total kinetic energy (TKE) (Simoncelli et al. (2011)) ratio between parent (MFS) and nested model (SANIFS). In our case this was calculated for the Gulf of Taranto both at the large and shelf-coastal scales.

Fig. 5a shows the TKE ratio between SANIFS and MFS as a function of the spin-up time calculated at the large scale. The impact of the hydrodynamics initialization is worth noting: SANIFS-v1 (red line) develops a higher kinetic energy than MFS, whereas SANIFS-v0 (blue line) is less energetic than MFS. Fig. 5a also shows that the steady condition (curve plateau or

at least a lower gradient of curve) for both experiments is reached after two days and kept quasi-constant on the subsequent spin-up days. At the shelf coastal scale (Fig. 5b), the TKE ratio for different spin-up days has the same qualitative behaviour as the one at the large scale.

In order to understand which model initialization fields are required, the model output was compared with the Sea Surface Temperature (SST) observations from satellites in the Taranto Gulf. Fig. 5c shows the root mean square error (RMSE) statistics

as a function of spin-up time for SANIFS-v0 and SANIFS-v1. For both model configurations, there is a decrease in RMSE after two days, but SANIFS-v1 is better than SANIFS-v0.

Finally, a further experiment in forecast mode was performed focusing on the coastal harbour scale (Mar Grande of Taranto) and compared with hourly velocity measurements at station P1 (Fig. 4). Fig. 5d confirms that SANIFS-v1 initialization has the lowest BIAS error.

A spin-up time of 3 days appears a reasonable choice to ensure the development of a own dynamics by the nested model. This choice were adopted also by other authors implementing high-resolution models in re-initialized mode both at large and coastal scale (Rolinski and Umgiesser (2005); Cucco et al. (2012); Trotta et al. (2016)) and also at harbour scale (Gaeta et al. (2016), in this issue). We conclude that a spin-up period of 3 days combined with active tracers and hydrodynamic initialization is the optimal choice for SANIFS initialization.

## 3.3  Forecast validation at large, shelf-coastal and coastal-harbour scales

This section investigates the SANIFS forecasting skills at a lead-time of one day in the large, shelf-coastal and coastal-harbour scales, using the MREA14 observations and in comparison with MFS. In order to assess the SANIFS forecasting skills using ECMWF atmospheric forcing, the following experiments, as reported in Fig. 6, were run: (i) FC-1, FC-2 and FC-3 for LS1 (1-3




October), (ii) FC-4 for CHS (5 October), (iii) FC-5 for SCS (8 October) and (iv) FC-6, FC-7 and FC-8 for LS2 (9-11 October). For each run 3 days of spin up define the initial conditions for the one day lead time forecast.

Fig. 7 refers to the SANIFS forecasting skills at LS1. In particular, Fig. 7a reports the comparison between modelled and observed representative profiles of temperature and salinity obtained averaging all the LS1 stations. The observed temperature profile is well reproduced by the model in the mixed layer, while the model thermocline is shifted upwards of about $10m$ in respect to the observed one, suggesting that future model investigations should be addressed to the improvement of the vertical mixing processes. For the salinity field, the higher discrepancies with the observations were found on the surface with a bias of 0.85 psu and could indicate the impact of atmospheric uncertainties of precipitation in parent model affecting the SANIFS initial condition. Figs. 7b-c show the SANIFS RMSE (red line with circles for temperature and green line with squares for salinity) and the RMSE Skill Score in respect with the reference model MFS forecasts (histrograms), calculated as:

$$SS_{RMSE,\phi} = 100 \frac{RMSE_{MFS,\phi} - RMSE_{SANIFS,\phi}}{RMSE_{MFS,\phi}} \quad [\%] \tag{1}$$

where $\phi$ indicates temperature and salinity. This expression identifies the percentage improvement (positive values) or worsening (negative values) of the SANIFS forecast in comparison with the MFS ones. The vertical average SANIFS RMSE is 0.55 °C for temperature and 0.18 psu for salinity.

Fig. 8a describes the representative profiles of temperature and salinity obtained averaging all the LS2 casts. In respect with the LS1, here SANIFS results are in better agreement with observations because the LS2 forecasts benefit of the LS1 data assimilation in MFS, thus reducing the overall error. Also for LS2 the SANIFS RMSE and the RMSE Skill Score in respect with the reference model MFS are shown (Figs. 8b-c). The vertical average SANIFS RMSE is 0.29 °C for temperature and 0.08 psu for salinity.

The assessment of RMSE skill score performed in the two large scale campaigns shows a slight improvement of SANIFS at the surface and mixing layer compared to MFS. Whereas the investigation reports a worsening result in the thermocline layers (between 40m and 55m for LS1, and 45m and 75m for LS2) likely due to vertical mixing issues.

Fig. 9a highlights the representative profiles of temperature and salinity obtained averaging all the SCS profiles. The observed temperature profile is well reproduced by the model in the mixed layer, while the modelled temperature gradient along the thermocline is less sharp than the observed one. This suggests that future model investigations should focus on the turbulence scheme and/or vertical discretization scheme of active tracers. The SANIFS RMSE and the RMSE Skill Score in respect with the reference model MFS are reported in Figs. 9b-c. The vertical average SANIFS RMSE is 0.59 °C for temperature and 0.13 psu for salinity.

In a comprehensive comparison with MFS at different scales, SANIFS forecasting skills result as follows: it is able to retain large scale dynamics of MFS and approaching to the shelf coastal scale to improve the forecast accuracy ($+17\%$ for temperature in Fig. 9b and $+5\%$ for salinity in Fig. 9c).

The numerical experiments mentioned above were repeated using COSMOME atmospheric forcing. The results on temperature and salinity fields (not shown) highlight no remarkable differences between the two configurations.





The added value of SANIFS can be further quantified at coastal harbour scale (CHS) which is not solved by the coarser resolution of MFS. Here comparisons in terms of sea temperature were carried out with the CTD collected on October 5 (see Fig. 4). Fig. 10a shows BIAS of temperature ($T_{mod} - T_{obs}$) at the surface ranging in ±0.25°C. Fig. 10b shows the RMSE profile and the vertical average RMSE is 0.11 °C. The vertical temperature structure is well captured by SANIFS, as seen

in Fig. 10cde, which shows three representative profiles at different depths. In detail, the model (i) keeps the temperature vertically constant from surface to bottom for the shallower bathymetry depth of 6 m (Fig. 10c, station 11), (ii) reproduces the temperature increase at the depth of 12m for station 12 (Fig. 10d), and (iii) simulates the temperature decrease at the deepest stations of the Mar Grande (20-25m, Fig. 10e).

### 3.4 Simulation tests at coastal harbour scale

In this section the SANIFS simulations are forced via MFS analysis and COSMOME analysis.

The model skills were evaluated in terms of the hourly sea temperature data recorded by the automatic monitoring station installed in Mar Grande (P1 in Fig. 4). Fig. 11 (bottom panel) shows the modelled hourly time-series of sea temperatures at 5 m of depth from the surface compared with the measured series. The mean absolute error calculated over all the hourly time steps is 0.13 °C. The time series can be split into three periods: up to 4 October, the mean temperature (about 23.4 °C) is

constant, then (from 4-8 October) it decreases (-0.5 °C), finally (from 8 to 11 October) is again constant (about 22.9 °C). The model simulates the daily cycle of temperature (as reported for instance from mid-day of 8 October to mid-day of 9 October) and complies with the 2m air temperature (T2M-COSMOME in top panel of Fig.11) used as forcing at surface. The highest difference in sea temperature between the model and the observed data is reported between 1 and 2 October (bottom panel of Fig.11). This corresponds to the highest discrepancy between T2M-COSMOME and the observed air temperature registered

at P1 station (top panel of Fig.11). The observed time-series of temperature seems to be affected by local atmospheric events such as rainfall. For instance, the two local minimum peaks of temperature (from 4 to 5 October and from 6 to 7 October) in observations may be due to the effect of total precipitation (blue histograms in Fig. 11 represent the COSMOME analysis of TP) since the maximum events of rainfall in the time series match with two local minimum temperatures. This suggests the need to introduce the temperature effects of rain (Gosnell et al. (1995)).

Fig. 12 compares the modeled and observed sea level at the tide-gauge station at P1 (Fig. 4). The tidal phases and amplitudes resulting from the model were estimated through a harmonic analysis of the sea surface elevation on an hourly basis using the TAPpy tidal analysis package (Cera (2011)). The results for the most important semidiurnal and diurnal constituents (M2, K1) are displayed in bottom left box in Fig. 12. The tidal amplitude range is about 20 cm. The percentage error was calculated for amplitude and phase, respectively as

$$E^{amp} = 100\frac{|A^o - A^m|}{A_o}, \quad E^{pha} = 100\frac{|P^o - P^m|}{180} \tag{2}$$

where $A$ and $P$ are the tidal amplitude and phase, and the superscripts $o$ and $m$ refer to observations and models, respectively. The tidal analysis reports errors of 6.2% and 14.1% for amplitude and phase of K1 component, and errors of 2.4% and 1.5%




for amplitude and phase of M2 component. The results appear satisfactory if compared with other reference studies for this area (Guarnieri et al. (2013); Ferrarin et al. (2013)).

Finally, velocity fields of SANIFS were compared with the observed ADCP data recorded at 5 m of depth from surface at station P1 in Fig. 13, which highlights a satisfactory model agreement with observations in terms of sea velocity direction (Fig. 13a) and an underestimation of the sea velocity intensity (bottom panel of Fig. 13b). This indicates that future investigations should focus on the turbulence scheme in coastal waters and/or bottom friction parametrization. Between 4 and 7 October, when the intensity of wind is higher (top panel of Fig. 13b with COSMOME and observed wind intensity at station P1), SANIFS is closer to the observed data in terms of sea current intensity. Furthermore the data confirm that sea velocity field is modulated by tidal signal (Scroccaro et al. (2004)) with semi-diurnal period (Malcangio and Mossa (2004)).

## 4 Circulation structures in the Gulf of Taranto

Fig. 14 presents the circulation patterns (at 30m of depth) in the Gulf of Taranto during the LS1 and LS2 phases of the MREA experiment. SANIFS describes a current circulation at the large scale characterized mainly by an anticyclonic structure (G1). At the first stage (LS1 survey, 1-3 October, Fig. 14a) the model produces weak cyclonically-oriented vortices in shelf-coastal areas (offshore of Gallipoli (V1), Taranto (V2) and Sibari (V3)). An intense coastal current (C1) strengthening the anticyclonic gyre offshore of Cape Santa Maria di Leuca impacts on the Adriatic coastal circulation producing a northward circulation along the southern Adriatic coasts. Although few observed experimental evidences of northward oriented WACC exists, this is one of the features of the SANIFS system over this period.

During the second cruise leg (SCS and LS2 surveys, 8-11 October, Fig. 14b) the large-scale anticyclonic gyre (G1) in the Gulf of Taranto becomes more intense (velocity peak of 0.2 m/s) and more extended, also covering the shelf-coastal areas and causing the three cyclonic vortices to vanish. The Gulf of Taranto circulation structure seems to affect the WACC (Artegiani et al. (1997a), Artegiani et al. (1997b); Cushman-Roisin et al. (2001)) entrance in the Gulf and along the Apulia coasts: in the case of a weaker G1 the WACC is reversed and it restarts when the G1 is stronger, isolating the Gulf of Taranto circulation from the rest of the domain.

Fig. 15 describes the daily mean surface and bottom circulation in Mar Grande for the 5 October (CHS survey). In the southeastern area of the basin, the circulation is characterized by an intensified current turning clockwise and developing a jet-like structure in the central area of Mar Grande. The two Mar Grande openings (Punta Rondinelle and Southern opening, Fig. 15a) show both a surface inflow to the semi-enclosed sea hinting to an antiestuarine dynamics mechanism (Fig. 15b describes a bottom outflow in the western part of the southern opening). De Pascalis et al. (2015) described the 2013 averaged fields of Mar Grande, showing that the basin is dominated by an estuarine dynamics but it could be that the Gulf would switch between the two opposite vertical circulations.



## 5 Conclusions

The SANIFS unstructured-grid forecasting system was developed to predict the three dimensional fields of active tracers and hydrodynamics for the Southern Adriatic Northern Ionian Seas, with a specific investigation for the Gulf of Taranto. The downscaling technique and the numerical settings adopted during the implementation phase make the system stable and robust, and allow short-time simulations at three different scales, from large to shelf-coastal, to coastal and harbour.

The forecast and simulation validation was performed by means of data collected during an experiment in October 2014.

Sensitivity tests on initialization procedures, focused on the assessment of the spin-up time and the choice of the dynamical fields to initialize the nested model, were carried out. The conclusion was that a spin-up period of 3 days combined with active tracers and hydrodynamic initialization is the optimal choice for initializing SANIFS.

The one day lead-time forecasts of SANIFS at the Gulf of Taranto scale, in open ocean, showed a vertical average RMSE of 0.55°C (LS1) and 0.29°C (LS2) for temperature and 0.18 psu (LS1) and 0.08 psu (LS2) for salinity. The maximum discrepancies were displayed at the thermocline for temperature and in surface for salinity.

The investigation at the different scales shows that SANIFS is able to correctly retain large scale dynamics of MFS and approaching to the shelf coastal scale to improve the forecast accuracy (+17% for temperature and +5% for salinity).

The strength of SANIFS was demonstrated at the coastal-harbour scale (CHS), where the system has a higher horizontal resolution (50 m in Mar Grande) and the error analysis showed a vertical average temperature RMSE of 0.11 °C.

Simulations underline the good performance of the system which reproduce (i) the daily cycle of temperature, (ii) the tidal amplitude and phase, and (iii) the velocity direction in Mar Grande. Moreover the error analysis hints to the need to introduce the temperature signal of rain and a detailed study of the vertical mixing parametrizations and bottom friction.

In the Gulf of Taranto SANIFS simulations describe a circulation mainly characterized by an anticyclonic gyre with the presence of cyclonic vortices in shelf-coastal areas. A water inflow from the open sea to Mar Grande characterizes both entrances to the semi-enclosed area.

SANIFS is under constant development and the numerical investigations in the future will focus on (i) turbulence scheme, (ii) parametrization of surface boundary conditions, (iii) initialization procedures based on fields with higher ageostrophical component, (iv) implementation of generalized Flather lateral boundary condition (Oddo and Pinardi (2008)), (v) introduction of data assimilation elements.

*Acknowledgements.* The research activities were funded by TESSA (PON01_02823/2) and IONIO (Subsidy Contract No. I1.22.05) projects. Partially funding through RITMARE (La Ricerca Italiana per il Mare) project is gratefully acknowledged. The authors wish to thank Dr. Georg Umgiesser (CNR-ISMAR) and Dr. Andrea Cucco (CNR-IAMC) for the scientific suggestions and tools provided during model implementation phase. The MREA14 team are thanked to provide observed data for validating the system.



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




| SANIFS (Southern Adriatic Northern Ionian coastal Forecasting System) configuration | |
|---|---|
| **Model** | SHYFEM (Umgiesser et al. (2004)) |
| **Horizontal resolution** | Open sea: $3 - 4$km. <br> Coastal area: $500 - 50$m. |
| **Vertical resolution** | Number of levels: 99. <br> The layer thickness is $2m$ until $40m$ from surface; <br> then progressively (stepwise) increased down to the bottom. |
| **Vertical mixing** | Pacanowski and Philander (1981) scheme modified by Lermusiaux (2001) |
| **Bottom friction** | Non-linear bottom parameterisation assuming a quadratic bottom friction. <br> Drag coefficient calculated by Maraldi et al. (2013). |
| **Initial Condition Fields** | Ocean: temperature, salinity, sea level and currents from MFS. |
| **Lateral Open Boundary Condition Fields** | Ocean: temperature, salinity, non-tidal sea level and total currents from MFS. <br> Tides: tidal elevation from OTPS. |
| **Rivers** | River runoffs: twenty monthly mean climatologies. |
| **Atmospheric forcing** | ECMWF and COSMOME. |
| **Spin-up time** | 3 days. |

**Table 1.** Configuration of SANIFS system.




**Figure 1.** SANIFS domain: horizontal grid overlapped on bathymetry for the whole domain.





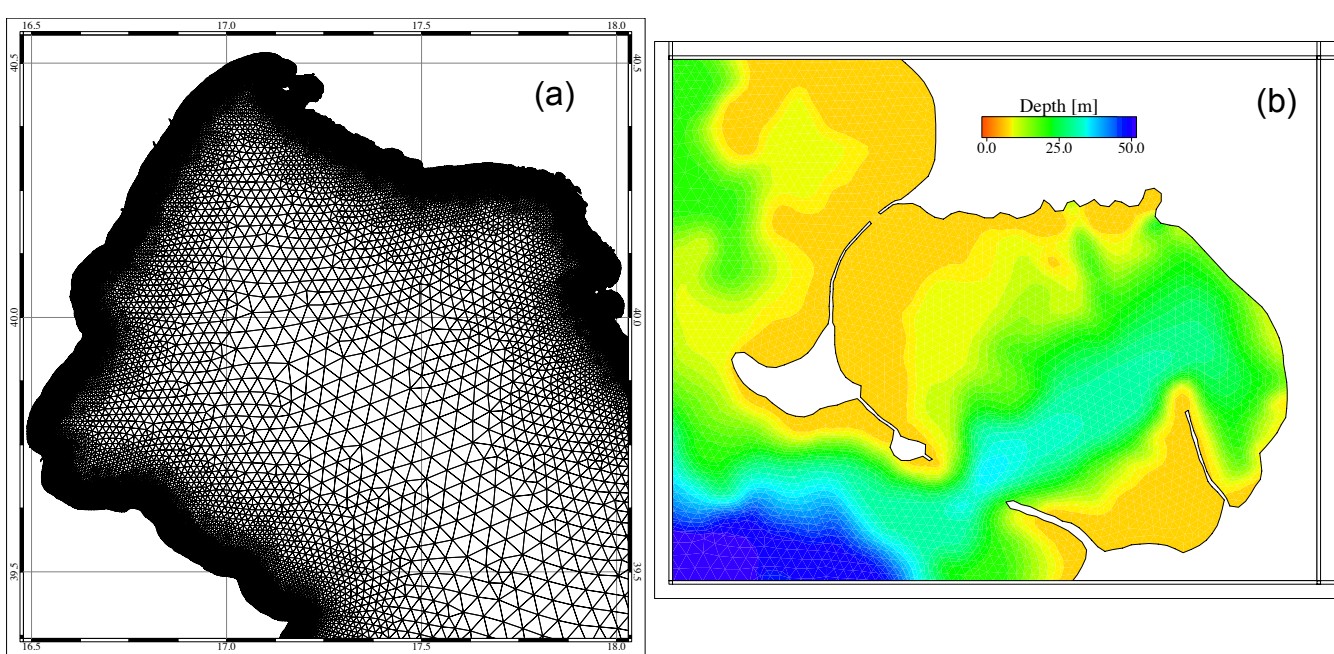

**Figure 2.** Enlarged views of horizontal grid Gulf of Taranto (a) and bathymetry in Mar Grande of Taranto (b).




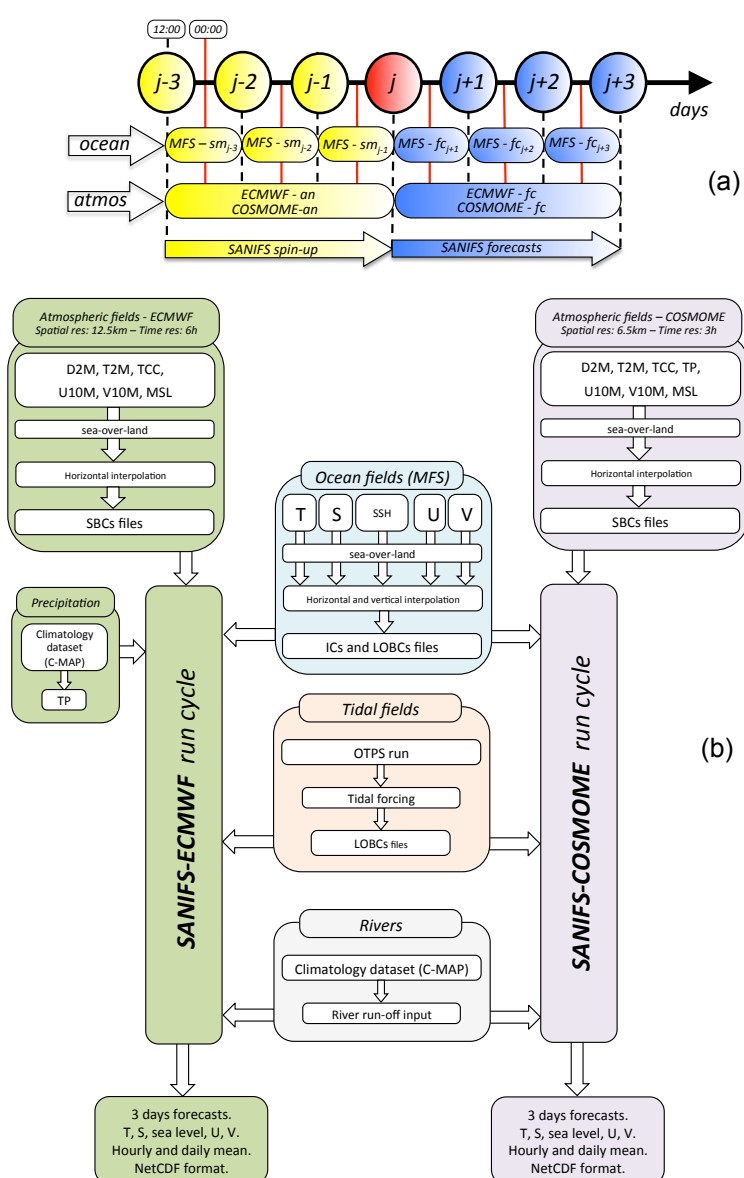

**Figure 3.** SANIFS daily forecast cycle (a) and operational chain (b).



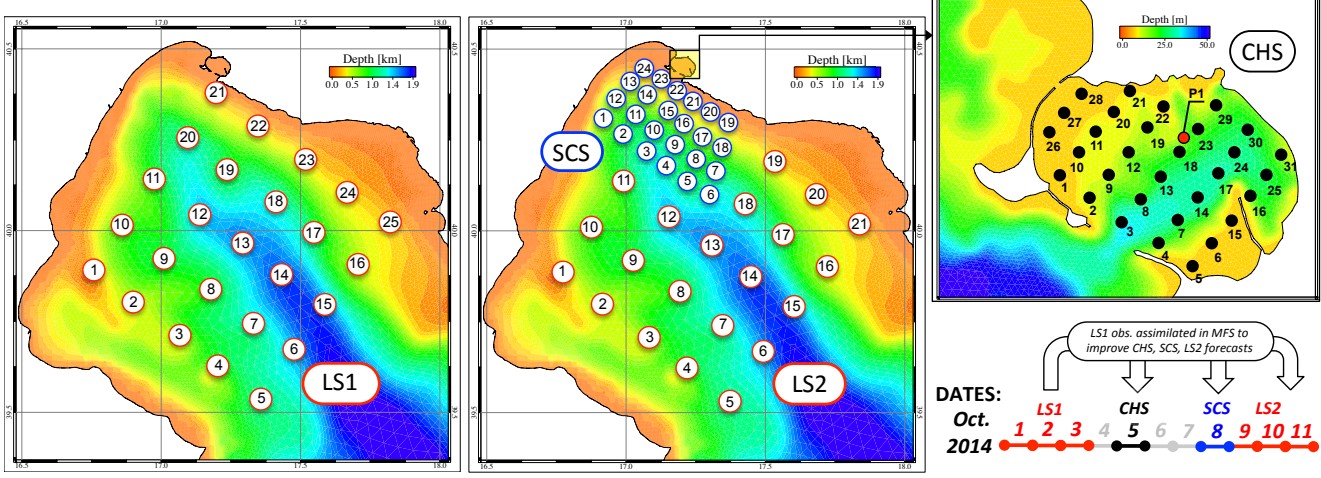

**Figure 4.** Sampling strategy of MREA2014 at the three scales. All the circle points refer to CTD stations. P1 refers to the fixed monitoring station in Mar Grande with CTD, ADCP and tidal measures. MREA timeline is also reported.






**Figure 5.** TKE ratio between downscaled model (SANIFS) and parent model (MFS) as function of spin-up time, calculated at large scale LS (a) and shelf-coastal scale SCS (b). Temperature RMSE (model vs. satellite SST) in the Gulf of Taranto as function of spin-up time (c). Velocity BIAS (m/s) calculated against hourly velocities measured at station P1 (Figure 4) in Mar Grande of Taranto (d). SANIFS-v0 refers to the experiment initialized through tracer fields. SANIFS-v1 refers to the experiment initialized through tracer and hydrodynamic fields.





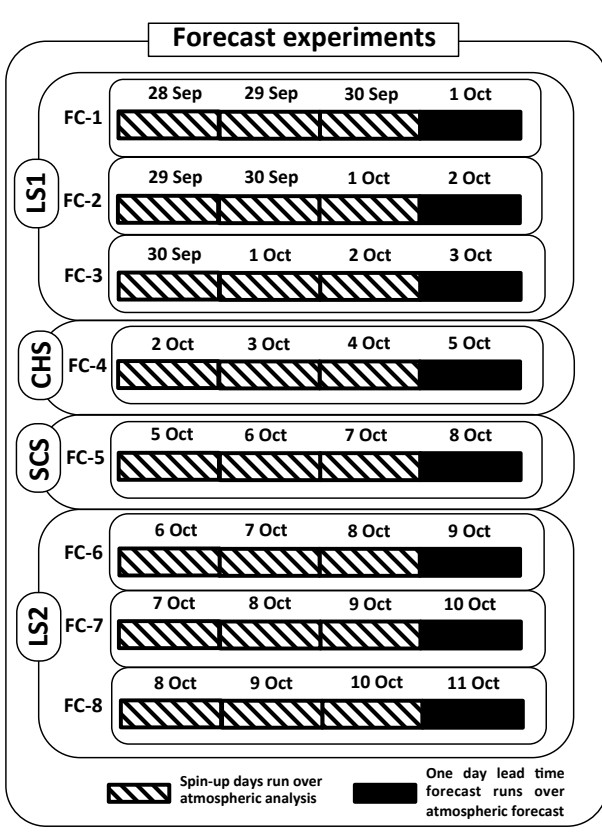

**Figure 6.** Sketch of forecast experiments performed at the different scales. The 3 days of spin up define the initial conditions for the one-day lead-time forecast.







**Figure 7.** SANIFS average profiles (temperature and salinity) compared with the observed ones for the LS1 (a). Temperature RMSE for SANIFS (red line with circles) and RMSE Skill Score compared with reference model MFS for the LS1 (b). Salinity RMSE for SANIFS (green line with squares) and RMSE Skill Score compared with reference model MFS for the LS1 (c). RMSE Skill Score is represented by histograms (positive values highlight levels where SANIFS produces more accurate forecasts than MFS; on the contrary negative values show the opposite).







**Figure 8.** SANIFS average profiles (temperature and salinity) compared with the observed ones for the LS2 (a). Temperature RMSE for SANIFS (red line with circles) and RMSE Skill Score compared with reference model MFS for the LS2 (b). Salinity RMSE for SANIFS (green line with squares) and RMSE Skill Score compared with reference model MFS for the LS2 (c). RMSE Skill Score is represented by histograms (positive values highlight levels where SANIFS produces more accurate forecasts than MFS; on the contrary negative values show the opposite).





**Figure 9.** SANIFS average profiles (temperature and salinity) compared with the observed ones for the SCS (a). Temperature RMSE for SANIFS (red line with circles) and RMSE Skill Score compared with reference model MFS for the SCS (b). Salinity RMSE for SANIFS (green line with squares) and RMSE Skill Score compared with reference model MFS for the SCS (c). RMSE Skill Score is represented by histograms (positive values highlight levels where SANIFS produces more accurate forecasts than MFS; on the contrary negative values show the opposite).





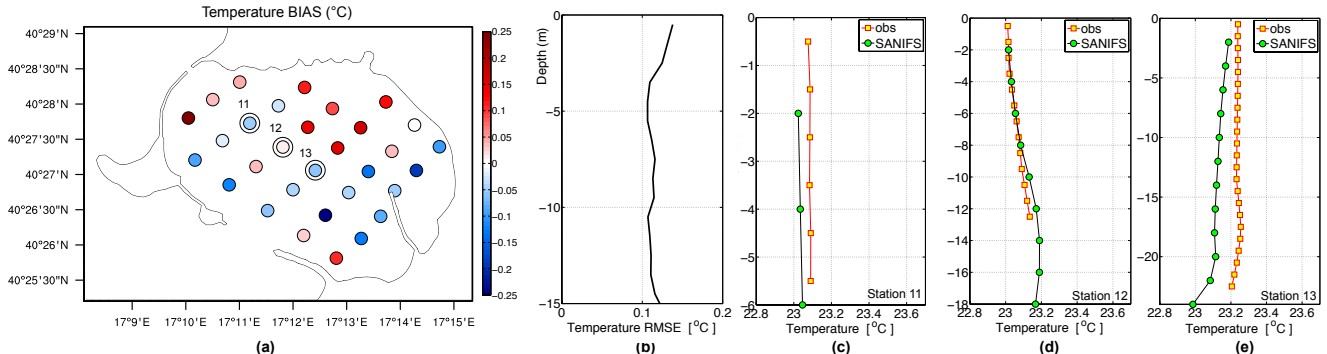

**Figure 10.** BIAS of surface temperature (a) and RMSE profile (b) for CHS campaign. Comparisons between modelled (green circles) and observed (yellow squares) profiles for casts 11 (c), 12 (d) and 13 (e).

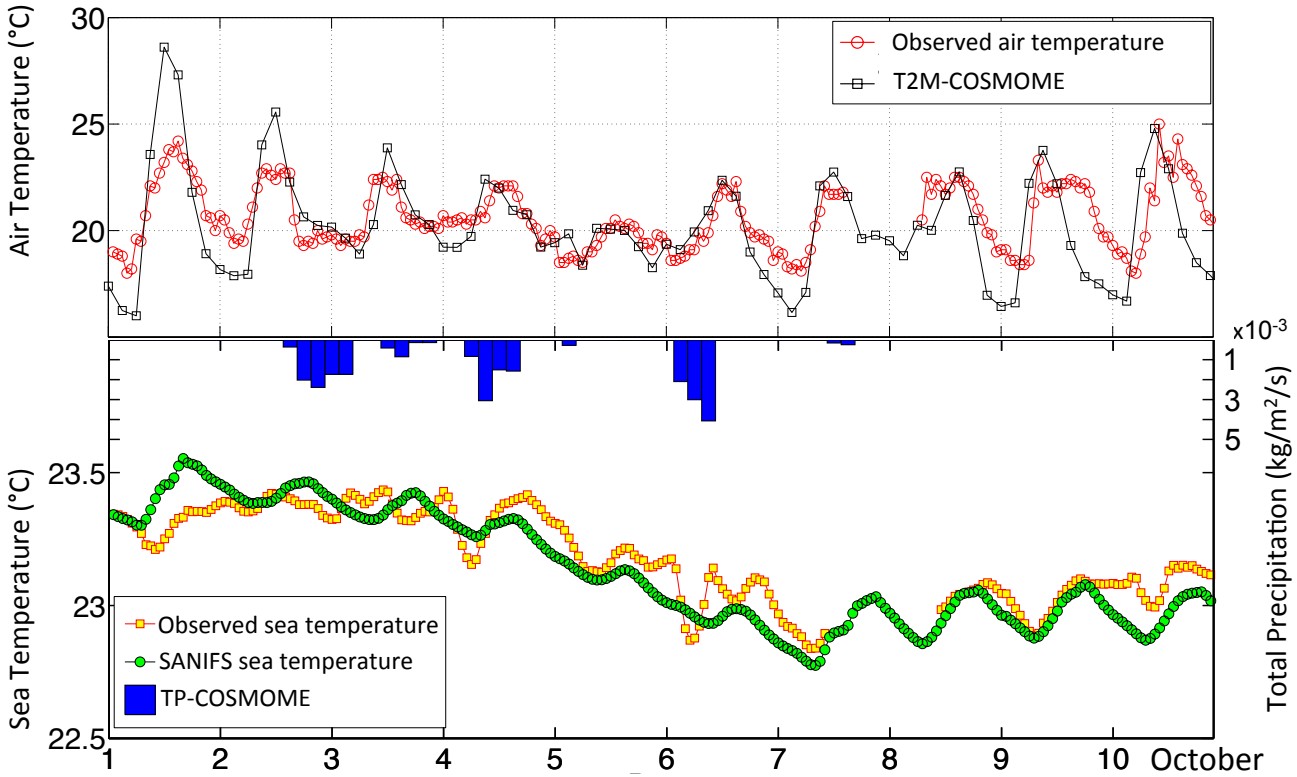

**Figure 11.** Bottom panel: hourly time-series of sea temperatures at a 5 m depth from the surface (green circles) compared with the measured series (yellow squares) for P1 station; blue histograms represent the COSMOME analysis of TP. Top panel: time-series of observed air temperature at P1 station (red circles) and 2 meter air temperature of COSMOME (black squares).





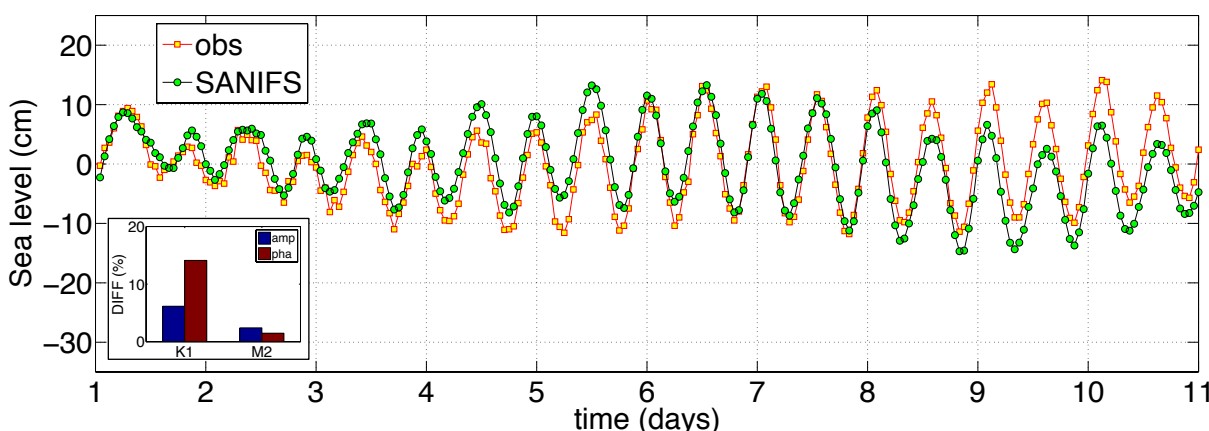

**Figure 12.** Hourly time-series of sea level modelled by SANIFS (green circles) and measured by P1 station (yellow squares). The box in the bottom left reports the tidal analysis for the most significant components (K1 and M2) in terms of amplitude and phase.







**Figure 13.** (a) Hourly time-series of observed (red rows) and modelled (black rows) sea velocity direction. (b) Hourly time-series of observed (yellow circles) and modelled (green squares) sea velocity intensity (bottom panel); time-series of observed (red circles) wind intensity at station P1 and 10m wind speed of COSMOME (black squares) (top panel).




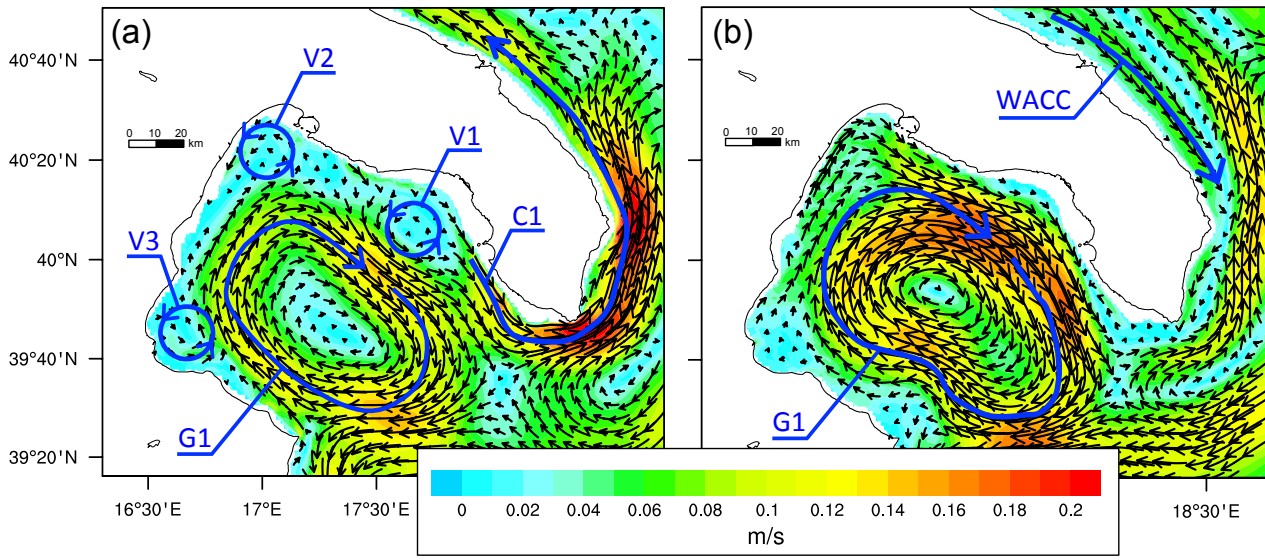

**Figure 14.** General circulation of SANIFS in Gulf of Taranto for (a) 1-3 October 2014 (LS2) and (b) 8-11 October 2014 (SCS and LS2).

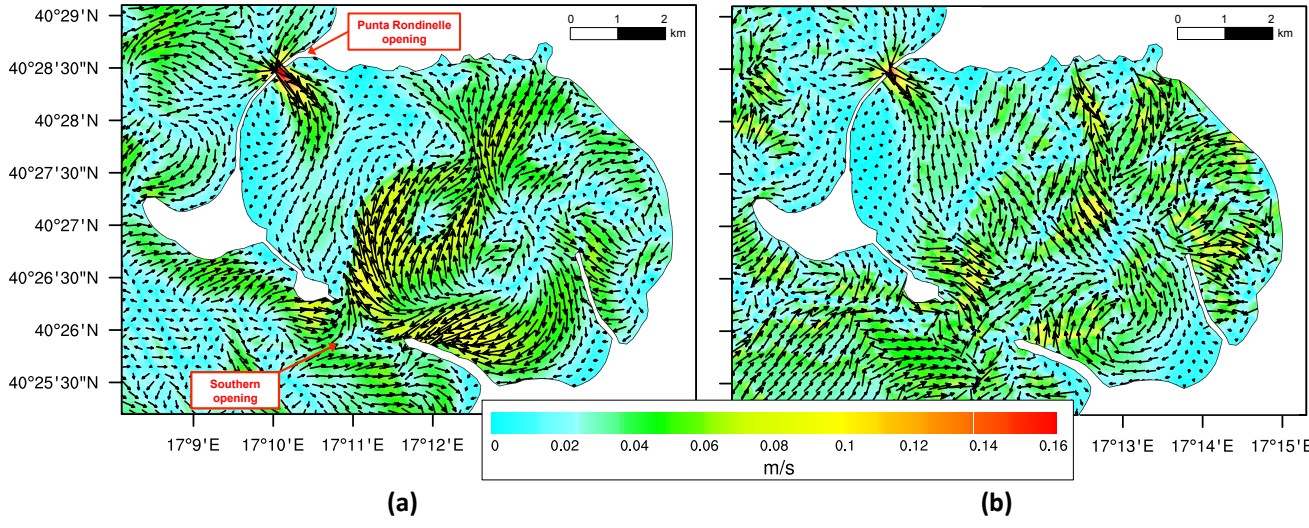

**Figure 15.** Surface (a) and bottom (b) circulation of SANIFS in Mar Grande of Taranto for 5 October 2014 (CHS).