# Peer review of "Coastal ocean forecasting with an unstructured-grid model in the Southern Adriatic Northern Ionian Sea"

_Natural Hazards and Earth System Sciences, 2016_

## Referee Comment (RC1) · Anonymous Referee #1 · 5 Aug 2016

nhess-2016-169

Coastal ocean forecasting with an unstructured-grid model in the Southern Adriatic Northern Ionian Sea Authors: I. Federico, N. Pinardi, G. Coppini, P. Oddo, R. Lecci, and M. Mossa Manuscript Type: Research article Special Issue: Situational sea awareness technologies for maritime safety and marine environment protection

General comments

In the present paper the SANIFS (Southern Adriatic Northern Ionian coastal Forecasting System) unstructured-grid forecast-ing system is adopted to predict the three dimensional fields of active tracers and hydrodynamics for the Southern Adriatic Northern Ionian Seas, with a specific investigation for the Gulf of Taranto. The downscaling technique and the numerical settings adopted during the implementation phase make the system stable and robust, and allow short-time simulations at three different scales, from large to shelf-coastal, to coastal and harbour. During the verification phase, SAN-IFS simulations are favourably compared with hourly time-series of temperature, sea level and velocity measured at the coastal-harbour scale showing a good agreement. First of all, the specific needs of each development stage are identified, and based on the state of the art. The paper is properly organized and concise. It is written clearly using correct grammar and syntax. The title is informative reflecting its contents. Scientific approach, methodology and work program are well outlined and also intelligible from the Abstract. The referenc-ing includes well sounded contributions from literature. It has been very interesting to read and review this paper, since I found it clear and useful for future works. I think that it meets international quality standards.

Specific comments

Pag. 5, lines 5-6: "River inflow surface salinity values were fixed to a constant value of 15 psu next to the river mouths, fol-lowing the sensitivity tests carried out with MFS parent model in the shelf areas close to river outlets". Please, specify how the value of 15 psu has been calibrated and/or add reference.

Pag 7, line 13: please substitute "hydrodynamics" with "hydrodynamic".

Pag. 10, lines 5-6: ". . . an underestimation of the sea velocity intensity (bottom panel of Fig. 13b). This indicates that future investigations should focus on the turbulence scheme in coastal waters and/or bottom friction parameterization". In the shal-low area the underestimation could be also related to currents induced by waves, not modelled by the system.

Pag. 10, lines 16-17: "Although few observed experimental evidences of northward oriented WACC exists, this is one of the features of the SANIFS system over this pe-riod". Could be appropriate to add reference of the mentioned "observed experi-mental

evidences". In addition, please substitute "exists" with "exist".

Further comments

The paper is focused on an innovative approach to operational oceanography, which consists in the use of unstructured grid model in forecast mode. The model products of this operational system can be adopted for many downstream fields, from applications to decisions support system and even to the field of coastal engineering.

Please also note the supplement to this comment:
http://www.nat-hazards-earth-syst-sci-discuss.net/nhess-2016-169/nhess-2016-169-RC1-supplement.pdf

---

## Referee Comment (RC2) · Anonymous Referee #2 · 19 Aug 2016

This well-written paper describes a new model system of the southern Adriatic and northern Ionian Seas, focusing on the Gulf of Taranto and Mar Grande of Taranto. The model produces a 3 day forecast of physical ocean properties in high spatial resolution. The focus of the paper is a validation of the modelled temperature, salinity, sea level and currents for a 1 day forecast. The validation is limited to one measuring campaign. However, this campaign seems to be fit for purpose. Operational ocean modelling is important for forecasting natural hazards on the sea and in the coastal region, and the topic is thus highly relevant for the journal. However, these issues could be discussed in more detail, as suggested below.

General questions / request for comments:

1) Considering the topic of the journal, please comment on the importance of the forecasts in relation to natural hazards of the area? Storm surges?

2) The model is reanalysed every day. I would expect a continued simulation from the day before to run equally well and possible allow more detailed coastal phenomena to build up, while saving computer resources. Please discus this option. I assume the arguments could be better assimilation in larger model domain? Challenges at the model boundary? Drift/instabilities of the model system?

3) What do the authors consider to be the smallest meaningful spatial resolution of the model? Hydrostatic approximations assume that the depth is much smaller than the length scales considered.

General notes

4) In some points below, I suggest to discuss various aspects of the results. Please consider if these discussions are best placed in the results text or in a new discussion section.

5) Check font sizes on all figures

6) Figure 7a, 8a and 9a: please add curves for the MFS model

7) Check spacing between numbers and units, and whether these are in italics

8) Please check commas, especially commas are missing after conjunctive adverbs at the start of a sentence

Specific notes (P: page, l: line)

9) Title: please add dash ( – ) between "Southern Adriatic" and "Northern Ionian"

10) P1 l4: please capitalize "system"

11) P1 l6: please capitalize Eastern in "South-eastern"

12) P1 l6: please change "500-50" to "50-500"

13) P2, l9: consider to change "could" to "can"

14) P2 l28-29: I find the description "The Southern Adriatic sea extends approximately southward along the latitude of 42°N" unclear

15) P3, l11: consider deleting "at least"

16) P4, l3: please add references to the different bathymetries used, or state the data source.

17) P4, l6: please state the maximum layer thickness

18) P4, l8: no comma after "where"

19) P4, l10: Please consider moving the reference towards the end of the sentence

20) P4 l26-30: please capitalise first letter in each sentence

21) P5 l 17: add "." after "et al"

22) P5 l26: consider deleting "only"

23) P5 l29: the sentence is a bit unclear. I assume that you mean "including tidal components" and not "including tidal free components"?

24) P8 l3-9: Addition of MFS curves as suggested in my point 5) will highlight if the effects come from the MFS model. To me it also seems likely that the model would benefit from a lower salinity in the river runoff forcing (makes sense with higher resolution), and that the mixing is influenced by the addition of tides. Please discuss.

25) P8 l15-19: Please discuss the effects of assimilation further. Which types of data are assimilated into the MFS system, how important is this for the model system, and how frequent are data available? This leads to the questions: Are the validation results of LS2 representative of the "normal" operating mode of the model, or is it more "normal" not to have data for assimilation as in LS1? How important would it be to have a more permanent source of observations?

26) P8 l23-28: is the effect of assimilation of LS1 into MFS also included here?

27) P9, l7: the feature (iii) is very small in the observations. Is it reported in other studies?

28) P9, l 24: Or, could it be that the rain is accompanied by increased winds/waves that could introduce upwelling, mixing or advection of colder water? The temperature drop seems to come in the beginning of the rainy periods.

29) P10, l2: It seems there is a (non-tidal?) sea level signal, causing sea level deviations of 5-10 cm. Comments?

30) P10, l12: Please add reference to Fig. 14 at "(G1)"

31) P10, l27-28: Please revise sentence and re-place parentheses

32) P11, l3: Please add dash ( – ) between "Southern Adriatic" and "Northern Ionian"

33) P11, l11: Please add that the difference is due to assimilation

34) P11, l19: "signal of rain": see note 28) and revise accordingly

35) P12, l23: remove comma after 1869

36) P12, l26: add dot after pp

37) Table 1: consider changing "ECMWF and COSMOME" to " ECMWF or COS-MOME"

38) Figure 1: is it possible to insert a small overview map (e.g. in the top right corner, covering e.g. 36-46N, 12-22E). Also, please mark Gulf of Taranto and Mar Grande of Taranto

39) Figure 2b: add latitude-longitude, or mark domain on figure 2a

---

## Author Comment (AC1) · 30 Nov 2016

In the following we have listed the referee comments and immediately after our response, with an explicit reference to the insertion of the revised texts and figures.

Furthermore, the revised version of manuscript is uploaded as supplement: we have highlighted the modifications suggested by Anonymous Referees #1 and #2 in blue and red, respectively.

For the specific notes, *P* means page and *l* line.

**Referee #1**

General comments

In the present paper the SANIFS (Southern Adriatic Northern Ionian coastal Forecasting System) unstructured-grid forecasting system is adopted to predict the three dimensional fields of active tracers and hydrodynamics for the Southern Adriatic Northern Ionian Seas, with a specific investigation for the Gulf of Taranto. The downscaling technique and the numerical settings adopted during the implementation phase make the system stable and robust, and allow short-time simulations at three different scales, from large to shelf-coastal, to coastal and harbour. During the verification phase, SANIFS simulations are favourably compared with hourly time-series of temperature, sea level and velocity measured at the coastal-harbour scale showing a good agreement.

First of all, the specific needs of each development stage are identified, and based on the state of the art. The paper is properly organized and concise. It is written clearly using correct grammar and syntax. The title is informative reflecting its contents. Scientific approach, methodology and work program are well outlined and also intelligible from the Abstract. The referencing includes well sounded contributions from literature.

It has been very interesting to read and review this paper, since I found it clear and useful for future works. I think that it meets international quality standards.

**Authors**

We thank the referee for his/her appreciation of our work.

**Referee #1**

Pag. 5, lines 5-6: "River inflow surface salinity values were fixed to a constant value of 15 psu next to the river mouths, following the sensitivity tests carried out with MFS parent model in the shelf areas close to river outlets". Please, specify how the value of 15 psu has been calibrated and/or add reference.

**Authors**

We have modified the sentence (*P*.5 – *l*.10-13) to further clarify this issue adding references. Here follows:

"*River inflow surface salinity values were fixed to a constant value of 15 psu next to the river mouths, following the sensitivity tests carried out with MFS parent model and the result of sensitivity tests performed by Simoncelli et al. (2011) on the basis of salinity profiles measured in the shelf areas close to river outlets. This value has been also adopted in other studies on the Adriatic circulation giving a realistic salinity profile for the open sea (Oddo et al., 2005).*"

**Referee #1**

Pag 7, line 13: please substitute "hydrodynamics" with "hydrodynamic".

**Authors**

Done.

**Referee #1**

Pag. 10, lines 5-6: "…an underestimation of the sea velocity intensity (bottom panel of Fig. 13b). This indicates that future investigations should focus on the turbulence scheme in coastal waters and/or bottom friction parameterization". In the shallow area the underestimation could be also related to currents induced by waves, not modelled by the system.

**Authors**

We agree with the referee. We have added a sentence (*P*.10 – *l*.17-19) related to this issue and a reference. Here follows:

"*…an underestimation of the sea velocity intensity (bottom panel of Fig. 13b). This indicates that future investigations should focus on the turbulence scheme in coastal waters and/or bottom friction parameterization. Furthermore, in the shallow area the underestimation could be also related to currents induced by waves (e.g. Gaeta et al., 2016), not modelled by SANIFS system.*"

**Referee #1**

Pag. 10, lines 16-17: "Although few observed experimental evidences of northward oriented WACC exists, this is one of the features of the SANIFS system over this period". Could be appropriate to add reference of the mentioned "observed experimental evidences". In addition, please substitute "exists" with "exist".

**Authors**

We modify the sentence adding references (*P*.10 – *l*.30-32). Here follows:

*"The few observed evidences of the northward oriented WACC are mainly due to upwelling favorable winds along the coast (Kourafalou, 1999; Rizzoli and Bergamasco, 1983)."*

**Referee #1**
Further comments
The paper is focused on an innovative approach to operational oceanography, which consists in the use of unstructured grid model in forecast mode. The model products of this operational system can be adopted for many downstream fields, from applications to decisions support system and even to the field of coastal engineering.
**Authors**
We thank the referee for his/her appreciation of our work.

**Referee #2**
This well-written paper describes a new model system of the southern Adriatic and northern Ionian Seas, focusing on the Gulf of Taranto and Mar Grande of Taranto. The model produces a 3 day forecast of physical ocean properties in high spatial resolution.
The focus of the paper is a validation of the modelled temperature, salinity, sea level and currents for a 1 day forecast. The validation is limited to one measuring campaign.
However, this campaign seems to be fit for purpose. Operational ocean modelling is important for forecasting natural hazards on the sea and in the coastal region, and the topic is thus highly relevant for the journal. However, these issues could be discussed in more detail, as suggested below.
**Authors**
We thank the referee for his/her appreciation of our work.

**Referee #2**
General questions / request for comments:
1) Considering the topic of the journal, please comment on the importance of the forecasts in relation to natural hazards of the area? Storm surges?
**Authors**
Thanks for the suggestion. We have added a sentence (*P*.2 – *l*.8-11) to enforce this point. Here follows:
"In particular, *a high-resolution operational forecasting system could contribute to take decisions for mitigation of natural hazards in coastal areas, such as storm surge events, minimizing their potential negative impacts on a wide range of coastal and maritime facilities, and reducing the damages to coastal communities.*"

**Referee #2**
2) The model is reanalysed every day. I would expect a continued simulation from the day before to run equally well and possible allow more detailed coastal phenomena to build up, while saving computer resources. Please discus this option. I assume the arguments could be better assimilation in larger model domain? Challenges at the model boundary? Drift/instabilities of

the model system?

**Authors**

The choice to adopt a reinitialized model is mainly due to two reasons:

a) The parent model (MFS-CMEMS http://marine.copernicus.eu/) is provided by data assimilation. Since the implementation strategy of the coastal operational system is to design a forecasting platform taking into account also the deep ocean fields propagating in coastal areas, the initialization by high quality large-scale fields (from a consolidated model - NEMO - enclosed in the framework of MFS with data assimilation component) impacts positively on the nested model forecasts.

b) An operational approach based on continuum simulation needs a long-term retrospective simulation (e.g. 10 years) to initialize the day-zero of the forecasting system. Currently, a frontier-going-to-state-of-art model based on fully 3D unstructured grid approach doesn't allow the above-mentioned long term run. This is mainly due to (i) computational time (please take into account that currently the SHYFEM model is in serial mode, but MPI parallelization activities are in progress) and (ii) potential instabilities and drifting by the model, to be verified with a long term run.

However the re-initialization methodology has been validated evaluating the spin-up time needed for a dynamical adjustment after the initialization from the interpolation of coarser ocean model fields (par. 3.2). The kinetic energy investigation shows that the assumption of 3-days spin up appears a reasonable choice to ensure the development of an own dynamics by the nested model. This choice was adopted also by other authors implementing high-resolution models in re-initialized mode both at large and coastal scale (Rolinski and Umgiesser (2005); Cucco et al. (2012); Trotta et al. (2016)) and also at harbour scale (Gaeta et al. (2016)).

In order to clarify the two above-mentioned issues [a) and b)] we have added two sentences in the text:

1) the first one in section (*P*.3 – *l*.21-25) "2. The forecasting system: definition and implementation" reads:

"*The re-initialization strategy allows exploiting the systematic and high-quality fields of parent model MFS (provided by data assimilation), which supplies operational forecasting products in the framework of CMEMS service (Copernicus Marine Environmental Monitoring Service, http://marine.copernicus.eu/). This type of approach has been adopted by other forecasting systems downscaled from MFS, as reported in Napolitano et al. (2016).*"

2) the second one in Conclusion section with an indication on the future developments that reads (*P*.12 – *l*.7-9):

"*… (vi) possibility to switch the operational chain from the every-day-reinitilized fields resulting from MFS system (currently adopted) to a continued-simulation approach starting every day from the initial conditions resulting from the SANIFS hindcast of the previous day.*"

**Referee #2**

3) What do the authors consider to be the smallest meaningful spatial resolution of the model? Hydrostatic approximations assume that the depth is

much smaller than the length scales considered.

**Authors**

The three meaningful resolutions of the model are 3-4 km in open-sea, 500 m in coastal area and 50 m in harbor scale (Mar Grande).

The grid has been designed to take into account the hydrostatic approximation. Nevertheless, there are some critical area (narrow shelves in Ionian Sea of Calabria) where the element resolution is 500m and the ratio between depth and length scale is close to the limit value for the hydrostatic approximation.

The reviewer suggestion could stimulate to use a consolidated non-hydrostatic unstructured model (currently SHYFEM is not equipped by non-hydrostatic core) assessing the improvement in shelf-coatal areas.

**Referee #2**

General notes

4) In some points below, I suggest to discuss various aspects of the results. Please consider if these discussions are best placed in the results text or in a new discussion section.

**Authors**

The same paper structure is kept; just we have named the section "Sensitivity tests, validation and discussion"

**Referee #2**

5) Check font sizes on all figures

**Authors**

We have checked the font sizes on all figures. The size has been increased for some figures to ensure readability.

**Referee #2**

6) Figure 7a, 8a and 9a: please add curves for the MFS model

**Authors**

In order to better appreciate the difference between parent model and nested, we have added BIAS sub-figure (now *b*) for the Figures 7, 8 and 9. This is also ensure the readability, differently from the overlapping between curves in subfigures *a*. The sub-figures 7b, 8b and 9b have been commented in the text (*P*. 8 – *l*.12-15; *P*. 8 – *l*.23-25; P9 – *l*.2-5)

**Referee #2**

7) Check spacing between numbers and units, and whether these are in italics

**Authors**

Done. All numbers-and-units have been standardized with space between numbers and units, and with italics for units.

**Referee #2**

8) Please check commas, especially commas are missing after conjunctive adverbs at the start of a sentence

**Authors**

Done.

**Referee #2**

Specific notes (P: page, l: line)

9) Title: please add dash ( – ) between "Southern Adriatic" and "Northern Ionian"

**Authors**

Done everywhere, also in the text.

**Referee #2**

10) P1 l4: please capitalize "system"

**Authors**

Done.

**Referee #2**

11) P1 l6: please capitalize Eastern in "South-eastern"

**Authors**

Done.

**Referee #2**

12) P1 l6: please change "500-50" to "50-500"

**Authors**

Done.

**Referee #2**

13) P2, l9: consider to change "could" to "can"

**Authors**

Done.

**Referee #2**

14) P2 l28-29: I find the description "The Southern Adriatic sea extends approximately southward along the latitude of 42 N" unclear

**Authors**

In literature, the Adriatic basin is subdivided 3 sub-basins (Northern, Middle and Southern, see e.g. Artegiani et al., 1998).

The Southern-Adriatic Sea extends from the Pelagosa sill (in open sea at latitude 42°N) to the Otranto Channel (at latitude 40°N).

We believe this description clarify the referee request.

**Referee #2**

15) P3, l11: consider deleting "at least"

**Authors**

Done.

**Referee #2**

16) P4, l3: please add references to the different bathymetries used, or state the data source.

**Authors**

We have stated the data source. The sentence (*P.4 – l.8-10*) changes in:
"*The SANIFS bathymetry (Fig. 1 and Fig. 2b) was derived from the U.S. Digital Bathymetric Data Base Variable Resolutions (DBDB-1) at 1' resolution for the Mediterranean basin and integrated with higher resolution bathymetry for coastal areas in the Gulf of Taranto provided by the Italian Navy Hydrographic Institute.*"

**Referee #2**

17) P4, l6: please state the maximum layer thickness

**Authors**

Done. We have added (*P.4 – l.13*): "*The maximum layer thickness at the bottom is 200m*".

**Referee #2**

18) P4, l8: no comma after "where"

**Authors**

Done.

**Referee #2**

19) P4, l10: Please consider moving the reference towards the end of the sentence

**Authors**

Done.

**Referee #2**

20) P4 l26-30: please capitalise first letter in each sentence

**Authors**

Done.

**Referee #2**

21) P5 l 17: add "." after "et al"

**Authors**

Done.

**Referee #2**

22) P5 l26: consider deleting "only"

**Authors**

Done.

**Referee #2**

23) P5 l29: the sentence is a bit unclear. I assume that you mean "including tidal components" and not "including tidal free components"?

**Authors**

The referee is correct. This was a typing mistake.

**Referee #2**

24) P8 l3-9: Addition of MFS curves as suggested in my point 5) will highlight

if the effects come from the MFS model. To me it also seems likely that the model would benefit from a lower salinity in the river runoff forcing (makes sense with higher resolution), and that the mixing is influenced by the addition of tides. Please discuss.

**Authors**

Thanks for the comment. We agree with the reviewer.

We have added a sentence for Fig. 8 (*P.8 – l.*24-25) where the improvement by SANIFS in salinity prediction for surface and mixing is better highlighted:

*"It is worth to note the lower BIAS of salinity for SANIFS in respect with MFS, which could indicate the impacts of river inputs (Crati, Bradano, Basento, Sinni and Agri)."*

Furthermore, a sentence (*P.8 – l.*20-21) on potential benefit of tides modelling for the mixing layer representation has been added:

*"The slight improvement of the SANIFS forecast in mixing layer representation could indicate the added value of tides modelling."*

**Referee #2**

25) P8 l15-19: Please discuss the effects of assimilation further. Which types of data are assimilated into the MFS system, how important is this for the model system, and how frequent are data available? This leads to the questions: Are the validation results of LS2 representative of the "normal" operating mode of the model, or is it more "normal" not to have data for assimilation as in LS1? How important would it be to have a more permanent source of observations?

**Authors**

We clarify this issue enforcing the text.

In parent system MFS, the real time data assimilation is based on the 3D variational scheme adapted to the oceanic assimilation problem (Dobricic and Pinardi, 2008). The assimilation cycle is daily.

The system assimilates both in-situ and satellite data, in particular:

(i) vertical profiles of temperature and salinity (Argo, XBT, Gliders) are assimilated from Copernicus InSitu-TAC (http://marine.copernicus.eu/, http://www.coriolis.eu.org/Data-Products/Data-Delivery/Copernicus-In-Situ-TAC)

(ii) the non-solar heat flux correction is achieved through satellite SST nudging,

(iii) along track Sea Level Anomaly are assimilated from Copernicus MSL-TAC, for all available satellites (Jason2, Cryosat2, Saral/AltiKa).

The data above mentioned are representative of the "normal" operating mode of the system. Looking at the in situ data, a lack of data for Gulf of Taranto is evident in the normal operating mode as highlighted by Figure *X* (vertical profiles of T/S from Argo, XBT and Gliders in October 2014).

Furthermore, the MREA oceanographic campaign has been specifically planned in the Gulf of Taranto because considered an area with a lack of data (Pinardi et al., 2016, a companion paper in this issue). In addition, the few observations reported in the past for this area have not a synoptic coverage.

The reducing of BIAS and RMSE from LS1 (with assimilation just in "normal"

operating mode) to LS2 (with assimilation in "normal" operating mode and assimilation of LS1 data) highlights how is important to have a more permanent source of observations with the synoptic coverage.

We have modified the sentence (*P.8 – l.25-28*) to enforce this question:

"*In respect with the LS1, here SANIFS results are in better agreement with observations because the LS2 forecasts benefit of the LS1 data assimilation in MFS, thus reducing the overall error. It is worth to note the higher forecast accuracy of both parent and nested models due to the assimilation of a source of observations with a synoptic coverage (Pinardi et al., 2016).*"

[Figure]

*Figure X: coverage of in-situ data in Northern Ionian Sea for October 2014 (the period of the oceanographic survey) from http://www.coriolis.eu.org/. No in-situ T/S profiles are present in Taranto Gulf area. This represents the "normal" operating mode of data assimilation component of MFS.*

**Referee #2**

26) P8 l23-28: is the effect of assimilation of LS1 into MFS also included here?

**Authors**

Yes, The effects of data assimilation of LS1 are included also for SCS as reported in the timeline of Figure 4. A sentence (*P.8 – l.6-7*) has been added to enforce this question:

"*The LS1 observations have been assimilated in MFS to improve forecasts of LS2, SCS and CHS.*"

**Referee #2**

27) P9, l7: the feature (iii) is very small in the observations. Is it reported in other studies?

**Authors**

The figure shows both general skills of model for the entire Mar Grande (Fig.

10ab) and specific features at different depths (Fig. 10cde). For what concerns Fig. 10e, the decrease of temperature (e.g. from 20m to 22.5m) is small both for observation and model. The aim of the Figure is to show a good fitting between observations and model.

**Referee #2**
28) P9, l 24: Or, could it be that the rain is accompanied by increased winds/waves that could introduce upwelling, mixing or advection of colder water? The temperature drop seems to come in the beginning of the rainy periods.
**Authors**
Thanks for the comment. We agree with the reviewer and we have modified the sentence (*P*.9/10 – *l*.33-34/1-2) to take into account the suggestion:
"*The observed time-series of temperature seems to be affected by local atmospheric events such as rainfall and large winds (see Fig. 13a). For instance, the effect of total precipitation (blue histograms in Fig. 11 represent the COSMOME analysis of TP) can contribute together with the increasing of localized wind effects to two local minimum peaks of temperature (from 4 to 5 October and from 6 to 7 October).*"

**Referee #2**
29) P10, l2: It seems there is a (non-tidal?) sea level signal, causing sea level deviations of 5-10 cm. Comments?
**Authors**
We agree about the deviations due to non-tidal sea level signal. We have added a sentence (*P*.10 – *l*.11-13) to mention it:
"*Although a non-tidal sea level signal causes deviations of 5-7 cm (e.g. 9-11 October), the results on tidal analysis appear satisfactory if compared with other reference studies for this area (Guarnieri et al. (2013); Ferrarin et al. (2013)).*"

**Referee #2**
30) P10, l12: Please add reference to Fig. 14 at "(G1)"
**Authors**
Reference (*P*.10 – *l*.26-27) has been added: (Pinardi et al., 2016).

**Referee #2**
31) P10, l27-28: Please revise sentence and re-place parentheses
**Authors**
The sentence has been revised and the parentheses replaced (*P*.11 – *l*.9-11):
"*The two Mar Grande openings (Punta Rondinelle and Southern opening) show both a surface inflow (Fig. 15a) to the semi-enclosed sea hinting to an antiestuarine dynamics mechanism, as reported in Fig. 15b where a bottom outflow in the western part of the southern opening is highlighted.*"

**Referee #2**
32) P11, l3: Please add dash ( – ) between "Southern Adriatic" and "Northern

Ionian"
**Authors**
Done.

**Referee #2**
33) P11, l11: Please add that the difference is due to assimilation
**Authors**
We have added a sentence (*P*.11 – *l*.24-25):
"For LS2 SANIFS results are in better agreement with observations because the LS2 forecasts benefit of the LS1 data assimilation in MFS."

**Referee #2**
34) P11, l19: "signal of rain": see note 28) and revise accordingly
**Authors**
We have revised the sentence without any speculation on signal of rain.

**Referee #2**
35) P12, l23: remove comma after 1869
**Authors**
Done.

**Referee #2**
36) P12, l26: add dot after pp
**Authors**
Done.

**Referee #2**
37) Table 1: consider changing "ECMWF and COSMOME" to " ECMWF or COSMOME"
**Authors**
Done.

**Referee #2**
38) Figure 1: is it possible to insert a small overview map (e.g. in the top right corner, covering e.g. 36-46N, 12-22E). Also, please mark Gulf of Taranto and Mar Grande of Taranto
**Authors**
Done.

**Referee #2**
39) Figure 2b: add latitude-longitude, or mark domain on figure 2a
**Authors**
Done.

[revised manuscript text omitted]

The paper is organized as follows. In Section 2 the design and implementation of the forecasting system are introduced. Section 3 reports the sensitivity experiments and the validation of the SANIFS system. Section 4 discusses the circulation structures emerging from the SANIFS system during the validation exercise, and concluding remarks are provided in Section 5.

**2  The forecasting system: definition and implementation**

The SANIFS methodology is based on the high resolution model re-initialization every day, similarly to the short term limited area atmospheric modelling practice (Mesinger et al. (1988)). The re-initialization strategy allows exploiting the systematic and high-quality fields of parent model MFS (provided by data assimilation), which supplies operational forecasting products in the framework of CMEMS service (Copernicus Marine Environmental Monitoring Service, http://marine.copernicus.eu/). This type of approach has been adopted by other forecasting systems downscaled from MFS, as reported in Napolitano et al. (2016).

In this section we report the main model settings, the surface boundary conditions, lateral boundary conditions and the operational configuration. Those are summarized in Table 1.

**2.1  Model settings**

The SANIFS forecasting system is based on the SHYFEM model which is a 3D finite element hydrodynamic model (Umgiesser et al. (2004); Cucco and Umgiesser (2006)) solving the Navier-Stokes equations by applying hydrostatic and Boussinesq approximations. The unstructured grid is Arakawa B with triangular meshes (Bellafiore and Umgiesser (2010); Ferrarin et al. (2013)), which provides an accurate description of irregular coastal boundaries.

The scalars are computed at grid nodes, whereas velocity vectors are calculated at the center of each element. Vertically a z-layer discretization is applied and most variables are computed in the center of each layer, whereas stress terms and vertical velocities are solved at the layer interfaces (Bellafiore and Umgiesser (2010)).

In the coastal waters of the eastern Italian coastlines, the model has a high spatial resolution, generally reaching an element size of 500m, with further refinements in specific areas (e.g. Mar Grande of Taranto, Fig. 2b) where the resolution reaches 50 $m$. In the open ocean areas, the horizontal resolution is approximately 3-4 $km$ with respect to the 6-7 $km$ of the parent model.

The SANIFS bathymetry (Fig. 1 and Fig. 2b) was derived from the U.S. Digital Bathymetric Data Base Variable Resolutions (DBDB-1) at 1' resolution for the Mediterranean basin and integrated with higher resolution bathymetry for coastal areas in the Gulf of Taranto provided by the Italian Navy Hydrographic Institute.

The vertical discretization has 99 levels. This is appropriate for solving the field both in coastal and open-sea areas. The vertical spacing is 2 $m$ until 40 $m$ from the sea surface, and the resolution is then progressively (stepwise) increased down to the bottom with a maximum layer thickness of 200 $m$.

A non-linear bottom parameterisation assuming a quadratic bottom friction was imposed. The friction coefficient was expressed as $R = C_D\sqrt{u^2 + v^2 + e_b}$ where $u$ and $v$ are the horizontal velocities, $C_D$ is the drag coefficient calculated by a logarithmic formulation (Maraldi et al. (2013)) and $e_b$ is the bottom turbulent kinetic energy due to tides, internal waves breaking and other short time scale currents. Following the Treguier's 1992 experiment and the MFS settings, $e_b$ was set to a value of $2.5 \cdot 10^{-3}\ m^2/s^2$ (Treguier (1992)) .

A local Richardson number dependent formulation was applied for the vertical momentum and tracer eddy coefficients with a specific constraint in the mixing layer (Lermusiaux (2001)). Using a scheme similar to Pacanowski and Philander (1981), the calculation of eddy viscosities and diffusivities are based on the Richardson number $Ri = N^2/(\partial\bar{U}/\partial z)^2$ where $N^2$ is Brunt-Vaisälä frequency and $\bar{U}(x,y)$ the velocity field.

If $Ri(x,y,z,t) \geq 0$, the eddy viscosity and diffusivity are set to $A_v = A_v^b + (v_0)/(1 + 5Ri)^2$ and $K_v = K_v^b + (v_0)/(1 + 5Ri)^3$. Otherwise if $Ri(x,y,z,t) < 0$ a convective adjustment is adopted ($A_v = 5 \cdot 10^{-3}\ m^2/s$ and $K_v = 5 \cdot 10^{-3}\ m^2/s$).

The background molecular coefficients are $A_v^b = 10^{-6}\ m^2/s$ and $K_v^b = 10^{-7}\ m^2/s$. The shear eddy viscosity is $v_0 = 5 \cdot 10^{-3}\ m^2/s$. An enhancement in the mixing layer (Lermusiaux (2001)) was adopted to transfer and dissipate the wind stress and the buoyancy fluxes. The vertical eddy coefficients within the Ekman depth $h_e(x,y,t)$ are set to empirical values calibrated for region and season $A_v^e = 1.5 \cdot 10^{-3}\ m^2/s$ and $K_v^e = 5 \cdot 10^{-4}\ m^2/s$). The Ekman depth is calculated as a function of turbulent friction velocity $u^*(x,y) = \sqrt{|\tau|/\rho_0}$ through the relationship $h_e = E_k u^*/f_0$, where $\tau$ is the wind stress vector, $\rho_0$ the reference density, $f_0$ is the Coriolis factor, and $E_k$ an empirical coefficient set to 0.7.

**2.2   Surface boundary conditions**

Four basic surface boundary conditions are used:

1. For temperature, the air-sea heat flux is parameterized by bulk formulas described in Pettenuzzo et al. (2010);

2. For momentum, surface stress is computed with the wind drag coefficient according to Hellermann and Rosenstein (1983);

3. For the free surface, a water flux is used containing evaporation minus precipitation and runoff;

4. For salinity, the turbulent salt flux is set equal to the product of the water flux and the surface salinity.

5    Twenty monthly mean climatological river runoffs (Verri et al. (2016)) were considered:

– Italian Ionian sea rivers: Basento, Bradano, Crati, Sinni, Agri, Neto;

– Italian Adriatic sea rivers: Fortore, Cervaro, Ofanto;

– Greek Ionian sea rivers: Arachthos, Thyamis;

– Albania-Montenegro-Croatia Adriatic sea rivers: Vijose, Seman, Shkumbi, Erzen, Ishm, Mat, Drin, Buna/Bojana, Neretva.

10    River inflow surface salinity values were fixed to a constant value of 15 $psu$ next to the river mouths, following the sensitivity tests carried out with MFS parent model and the result of sensitivity tests performed by Simoncelli et al. (2011) on the basis of salinity profiles measured in the shelf areas close to river outlets. This value has been also adopted in other studies on the Adriatic circulation giving a realistic salinity profile for the open sea (Oddo et al. (2005)).

[revised manuscript text omitted]

Fig. 7 refers to the SANIFS forecasting skills at LS1. In particular, Fig. 7a reports the comparison between modelled and observed representative profiles of temperature and salinity obtained averaging all the LS1 stations. The observed temperature profile is well reproduced by the model in the mixed layer, while the model thermocline is shifted upwards of about $10\ m$ in respect to the observed one, suggesting that future model investigations should be addressed to the improvement of the vertical mixing processes. Fig. 7b shows the BIAS of the average profiles compared with the observations. For the salinity field, the higher discrepancies with the observations were found on the surface with a bias of $0.80\ psu$ and could indicate the impact of atmospheric uncertainties of precipitation in parent model affecting the SANIFS initial condition. However this discrepancy in surface with observation is lower than the one between MFS and observations. Figs. 7c-d show the SANIFS RMSE (red line with circles for temperature and green line with squares for salinity) and the RMSE Skill Score in respect with the reference model MFS forecasts (histrograms), calculated as:

$$SS_{RMSE,\phi} = 100\ \frac{RMSE_{MFS,\phi} - RMSE_{SANIFS,\phi}}{RMSE_{MFS,\phi}}\quad [\%] \tag{1}$$

where $\phi$ indicates temperature and salinity. This expression identifies the percentage improvement (positive values) or worsening (negative values) of the SANIFS forecast in comparison with the MFS ones. The slight improvement of the SANIFS forecast in mixing layer representation could indicate the added value of tides modelling. The vertical average SANIFS RMSE is $0.55\ °C$ for temperature and $0.18\ psu$ for salinity.

Fig. 8a describes the representative profiles of temperature and salinity obtained averaging all the LS2 casts and Fig. 8b shows the BIAS with the observations. The lower BIAS of salinity for SANIFS in respect with MFS could indicate the impacts of river inputs (Crati, Bradano, Basento, Sinni and Agri). In respect with the LS1, here SANIFS results are in better agreement with observations because the LS2 forecasts benefit of the LS1 data assimilation in MFS, thus reducing the overall error. It is worth to note the higher forecast accuracy of both parent and nested models due to the assimilation of a source of observations with a synoptic coverage Pinardi et al. (2016). Also for LS2 the SANIFS RMSE and the RMSE Skill Score in respect with the reference model MFS are shown (Figs. 8c-d). The vertical average SANIFS RMSE is $0.29\ °C$ for temperature and $0.08\ psu$ for salinity.

The assessment of RMSE skill score performed in the two large scale campaigns shows a slight improvement of SANIFS at the surface and mixing layer compared to MFS. Whereas the investigation reports a worsening result in the thermocline layers (between $40\ m$ and $55\ m$ for LS1, and $45\ m$ and $75\ m$ for LS2) likely due to vertical mixing issues.

Fig. 9a highlights the representative profiles of temperature and salinity obtained averaging all the SCS profiles. The observed temperature profile is well reproduced by the model in the mixed layer. The modelled temperature gradient along the thermocline, despite is less sharp than the observed one, is in better agreement with observations in comparison with MFS, as reported in Fig. 9b where the BIAS of the two systems are highlighted. Future model investigations should focus on the turbulence scheme and/or vertical discretization scheme of active tracers to improve further the thermocline representation. The SANIFS RMSE and the RMSE Skill Score in respect with the reference model MFS are reported in Figs. 9c-d. The vertical average SANIFS RMSE is 0.59 °$C$ for temperature and 0.13$psu$ for salinity.

In a comprehensive comparison with MFS at different scales, SANIFS forecasting skills result as follows: it is able to retain large scale dynamics of MFS and approaching to the shelf coastal scale to improve the forecast accuracy (+17% for temperature in Fig. 9c and +5% for salinity in Fig. 9d).

The numerical experiments mentioned above were repeated using COSMOME atmospheric forcing. The results on temperature and salinity fields (not shown) highlight no remarkable differences between the two configurations.

The added value of SANIFS can be further quantified at coastal harbour scale (CHS) which is not solved by the coarser resolution of MFS. Here comparisons in terms of sea temperature were carried out with the CTD collected on October 5 (see Fig. 4). Fig. 10a shows BIAS of temperature ($T_{mod} - T_{obs}$) at the surface ranging in ±0.25 °$C$. Fig. 10b shows the RMSE profile and the vertical average RMSE is 0.11 °$C$. The vertical temperature structure is well captured by SANIFS, as seen in Fig. 10cde, which shows three representative profiles at different depths. In detail, the model (i) keeps the temperature vertically constant from surface to bottom for the shallower bathymetry depth of 6 $m$ (Fig. 10c, station 11), (ii) reproduces the temperature increase at the depth of 12 $m$ for station 12 (Fig. 10d), and (iii) simulates the temperature decrease at the deepest stations of the Mar Grande (20-25 $m$, Fig. 10e).

**3.4 Simulation tests at coastal harbour scale**

In this section the SANIFS simulations are forced via MFS analysis and COSMOME analysis.

The model skills were evaluated in terms of the hourly sea temperature data recorded by the automatic monitoring station installed in Mar Grande (P1 in Fig. 4). Fig. 11 (bottom panel) shows the modelled hourly time-series of sea temperatures at 5 $m$ of depth from the surface compared with the measured series. The mean absolute error calculated over all the hourly time steps is 0.13 °$C$. The time series can be split into three periods: up to 4 October, the mean temperature (about 23.4 °$C$) is constant, then (from 4-8 October) it decreases (-0.5 °$C$), finally (from 8 to 11 October) is again constant (about 22.9 °$C$). The model simulates the daily cycle of temperature (as reported for instance from mid-day of 8 October to mid-day of 9 October) and complies with the 2 $m$ air temperature (T2M-COSMOME in top panel of Fig.11) used as forcing at surface. The highest difference in sea temperature between the model and the observed data is reported between 1 and 2 October (bottom panel of Fig.11). This corresponds to the highest discrepancy between T2M-COSMOME and the observed air temperature registered at P1 station (top panel of Fig.11).

The observed time-series of temperature seems to be affected by local atmospheric events such as rainfall and large winds (see 13a). For instance, the effect of total precipitation (blue histograms Fig. 11 represent the COSMOME analysis of TP)

can contribute together with the increasing of localized wind effects to two local minimum peaks of temperature (from 4 to 5 October and from 6 to 7 October).

[revised manuscript text omitted]

SANIFS is under constant development and the numerical investigations in the future will focus on (i) turbulence scheme, (ii) parametrization of surface boundary conditions, (iii) initialization procedures based on fields with higher ageostrophical component, (iv) implementation of generalized Flather lateral boundary condition (Oddo and Pinardi (2008)), (v) introduction of data assimilation elements, (vi) possibility to switch the operational chain from the every-day-reinitilized fields resulting from MFS system (currently adopted) to a continued-simulation approach starting every day from the initial conditions resulting from the SANIFS hindcast of the previous day.

[revised manuscript text omitted]

Pettenuzzo, D., Large, W.G., and Pinardi, N.: On the corrections of ERA40 surface flux products consistent with the Mediterranean heat and

10    water budgets and the connection between basin surface total heat flux and NAO, Journal of Geophysical Research, 115, 2010.

Pinardi, N., Allen, I., Demirov, E., De Mey, P., Korres, G., Lascaratos, A., Le Traon, P.Y., Maillard, C., Manzella, G., and Tziavos, C.: The Mediterranean ocean Forecasting System: first phase of implementation (1998-2001), Annales Geophysicae, 21, 3-20, 2003.

Pinardi, N. and Coppini, G., Operational oceanography in the Mediterranean Sea: the second stage of development, Ocean Sci., 6, 263-267, 2010.

15    Pinardi, N., Lyubartsev, V., Cardellicchio, N., Caporale, C., Ciliberti, S., Coppini, G., De Pascalis, F., Dialti, L., Federico, I., Filippone, M., Grandi, A., Guideri, M., Lecci, R., Lamberti, L., Lorenzetti, G., Lusiani, P., Macripó, C.D., Maicu, F., Tartarini, D., Trotta, F., Umgiesser, G., and Zaggia L.; Marine Rapid Environment Assessment in the Gulf of Taranto: a multiscale approach, Nat. Hazard Earth Sys., submitted in this issue, 2016.

Poulain, P.M.: Adriatic Sea surface circulation as derived from drifter data between 1990 and 1999, J. Mar. Syst. 29, 3-32, 2001.

20    Rizzoli, P.M. and Bergamasco, A.: The Dynamics of the Coastal Region of the Northern Adriatic Sea, Journal Physical Oceanography, 13, 1105-1130, 1983.

[revised manuscript text omitted]

**Figure 7.** SANIFS average profiles (temperature and salinity) compared with the observed ones for the LS1 (a). BIAS with LS1 observations of the MFS and SANIFS average profiles (temperature and salinity) (b). Temperature RMSE for SANIFS (red line with circles) and RMSE Skill Score compared with reference model MFS for the LS1 (c). Salinity RMSE for SANIFS (green line with squares) and RMSE Skill Score compared with reference model MFS for the LS1 (d). RMSE Skill Score is represented by histograms (positive values highlight levels where SANIFS produces more accurate forecasts than MFS; on the contrary negative values show the opposite).

[Figure]

**Figure 8.** SANIFS average profiles (temperature and salinity) compared with the observed ones for the LS2 (a). BIAS with LS2 observations of the MFS and SANIFS average profiles (temperature and salinity) (b). Temperature RMSE for SANIFS (red line with circles) and RMSE Skill Score compared with reference model MFS for the LS2 (c). Salinity RMSE for SANIFS (green line with squares) and RMSE Skill Score compared with reference model MFS for the LS2 (d). RMSE Skill Score is represented by histograms (positive values highlight levels where SANIFS produces more accurate forecasts than MFS; on the contrary negative values show the opposite).

[Figure]

**Figure 9.** SANIFS average profiles (temperature and salinity) compared with the observed ones for the SCS (a). BIAS with SCS observations of the MFS and SANIFS average profiles (temperature and salinity) (b). Temperature RMSE for SANIFS (red line with circles) and RMSE Skill Score compared with reference model MFS for the SCS (c). Salinity RMSE for SANIFS (green line with squares) and RMSE Skill Score compared with reference model MFS for the SCS (d). RMSE Skill Score is represented by histograms (positive values highlight levels where SANIFS produces more accurate forecasts than MFS; on the contrary negative values show the opposite).

[Figure]

**Figure 10.** BIAS of surface temperature (a) and RMSE profile (b) for CHS campaign. Comparisons between modelled (green circles) and observed (yellow squares) profiles for casts 11 (c), 12 (d) and 13 (e).

**Figure 11.** Bottom panel: hourly time-series of sea temperatures at a 5 m depth from the surface (green circles) compared with the measured series (yellow squares) for P1 station; blue histograms represent the COSMOME analysis of TP. Top panel: time-series of observed air temperature at P1 station (red circles) and 2 meter air temperature of COSMOME (black squares).

[Figure]

**Figure 12.** Hourly time-series of sea level modelled by SANIFS (green circles) and measured by P1 station (yellow squares). The box in the bottom left reports the tidal analysis for the most significant components (K1 and M2) in terms of amplitude and phase.

[Figure]

**Figure 13.** (a) Hourly time-series of observed (red rows) and modelled (black rows) sea velocity direction. (b) Hourly time-series of observed (yellow circles) and modelled (green squares) sea velocity intensity (bottom panel); time-series of observed (red circles) wind intensity at station P1 and 10m wind speed of COSMOME (black squares) (top panel).

[Figure]

**Figure 14.** General circulation of SANIFS in Gulf of Taranto for (a) 1-3 October 2014 (LS2) and (b) 8-11 October 2014 (SCS and LS2).

[Figure]

**Figure 15.** Surface (a) and bottom (b) circulation of SANIFS in Mar Grande of Taranto for 5 October 2014 (CHS).